# Cyclo(l-Pro–l-Leu) of *Pseudomonas putida* MCCC 1A00316 Isolated from Antarctic Soil: Identification and Characterization of Activity against *Meloidogyne incognita*

**DOI:** 10.3390/molecules24040768

**Published:** 2019-02-20

**Authors:** Yile Zhai, Zongze Shao, Minmin Cai, Longyu Zheng, Guangyu Li, Ziniu Yu, Jibin Zhang

**Affiliations:** 1State Key Laboratory of Agricultural Microbiology and National Engineering Research Center of Microbe Pesticides, College of Life Science and Technology, Huazhong Agricultural University, Wuhan 430070, China; zhaiyile1991@163.com (Y.Z.); cmm114@mail.hzau.edu.cn (M.C.); ly.zheng@mail.hzau.edu.cn (L.Z.); yz41@mail.hzau.edu.cn (Z.Y.); 2Key Laboratory of Marine Biogenetic Resources, Third Institute of Oceanography, State Oceanic Administration, Xiamen 361005, China; shaozz@163.com (Z.S.); mccc_ligy@163.com (G.L.)

**Keywords:** *Pseudomonas putida* MCCC 1A00316, *Meloidogyne incognita*, cyclo(l-Pro–l-Leu), nematicidal activity

## Abstract

*Pseudomonas putida* MCCC 1A00316 was originally isolated from an Antarctic soil and has demonstrated potential nematicidal activity. Thus, it has promising applications for the biological control of *Meloidogyne incognita*. The larval mortality and egg-hatching inhibition rates of *M. incognita* will increase with the rising concentration of culture filtrates of *P. putida* MCCC 1A00316 and the duration of exposure. Thus, this study aimed to separate, purify, and identify nematicidal compounds from *P. putida* MCCC 1A00316 and to validate their anti-*M. incognita* activities. Compounds were purified through silica gel column chromatography and thin-layer chromatography combined with high-performance liquid chromatography (HPLC). Structural identification was conducted through liquid chromatography time-of-flight mass spectrometry, ^1^H nuclear magnetic resonance (NMR) spectroscopy, ^13^C-NMR, and Marfey’s method. The isolated compounds were identified as cyclo(l-Pro–l-Leu) on the basis of the results of the above analyses and previously reported data. The effects of various concentrations of cyclo(l-Pro–l-Leu) on the mortality rates of second-stage juveniles (J2) of *M. incognita* were investigated. Results showed that HPLC-purified cyclo(l-Pro–l-Leu) displayed nematicidal activities. The mortality rate of *M. incognita* J2 reached 84.3% after 72 h of exposure to 67.5 mg/L cyclo(l-Pro–l-Leu). The lowest egg-hatching rate (9.74%) was observed after 8 days of incubation with 2000 mg/L cyclo(l-Pro–l-Leu). An egg-hatching rate of 53.11% was obtained under the control treatment (sterile distilled water). However, cyclo(l-Pro–l-Leu) did not elicit chemotaxis activity to *M. incognita*. This is the first work to investigate the anti-*M. incognita* characteristics of cyclo(l-Pro–l-Leu).

## 1. Introduction

Phytoparasitic nematodes are ubiquitous microscopic soil pests that cause severe crop losses by parasitizing plant roots [1]. The root-knot nematode, *Meloidogyne incognita*, is one of the most important phytoparasitic nematode species; infections by this species induce root galling, prevent plant growth, and cause nutrient deficiency [2,3]. *M. incognita* infections result in yield losses of 25.6% [4]. The global crop losses caused by nematodes exceed $150 billion annually [5]. Therefore, the presence of this pest in cultivated crops must be controlled [6,7,8].

The most common methods for the control of root-knot nematodes include the use of resistant cultivars, crop rotation, and chemical nematicidal agents. Nevertheless, resistance to root-knot nematodes is unstable and often reduces yields [9], and crop rotation is not always practical. In addition, the overuse of chemical nematicidal agents may result in environmental contamination, exert deleterious effects on beneficial organisms, and promote the development of nematocide-insensitive strains [10]. Therefore, economical, effective, and environmentally friendly nematode control methods are urgently needed.

Biocontrol agents have attracted considerable attention in recent years because of their safety to the environment, humans, especially food and vegetable security and animals [11]. Various pathogenic bacterial and fungal species have been isolated for the biocontrol of phytoparasitic nematodes; most of these organisms can produce nematicidal compounds [12]. The nematicidal mode of action of *Pasteuria penetrans* is mainly based on the parasitization of target nematodes [13]. Sphingosine produced by *Bacillus cereus* S2 has demonstrated intense nematicidal activity against *M. incognita* [14]. Chaetoglobosin A, a nematicidal compound produced by *Chaetomium globosum*, can adversely affect the survival rates of juvenile *M. incognita* with an LC_50_ of 77 μg/mL [15]. 

*Pseudomonas putida* MCCC 1A00316 produces various nematicidal factors, such as protein alkaline metalloproteinase AprA; the secondary metabolites hydrogen cyanide and cyclo(l-Ile–l-Pro); and several volatile organic compounds. Guo et al. reported that 96 h of exposure to 500 mg/L cyclo(l-Ile–l-Pro) from *P. putida* resulted in a mortality rate of 50% [16]. Zhai et al. found that seven volatile organic compounds, including dimethyl–disulfide, 1-undecene, 2-nonanone, 2-octanone, (*Z*)-hexen-1-ol acetate, 2-undecanone, and 1-(ethenyloxy)octadecane, presented nematicidal activity against second-stage juveniles (J2) of *M. incognita* and inhibited the egg hatching of *M. incognita* when applied as a direct-contact nematocide and as a fumigant [17]. Numerous other nematicidal compounds require identification and study.

*Pseudomonas* species are ubiquitous in nature and produce a large number of secondary metabolites with activities against important plant diseases [18]. In the present study, we found that *P. putida* MCCC 1A00316 from an Antarctic soil showed nematicidal potential and thus has promising applications for the biological control of *M. incognita*. Therefore, we separated and purified potential nematicidal compounds from *P. putida* MCCC 1A00316. We isolated and identified one nematicidal compound and characterized its anti-*M. incognita* activity. 

## 2. Results

### 2.1. Nematicidal Activity of Culture Filtrates from P. Putida MCCC 1A00316

The nematicidal effects of culture filtrates from isolates of MCCC 1A00316 were evaluated through a direct-contact bioassay (Figure 1 and Figure 2). The egg-hatching of *M. incognita* was affected by exposure to the culture filtrates of MCCC 1A00316 in a dose-dependent manner. The emergence of J2 was inversely related to filtrate concentration and exposure duration (Figure 1). At 6 days after incubation, egg hatching was drastically inhibited under treatment with culture filtrates (from 1× to 6 ×) relative to that under treatment with sterile distilled water (SDW) treatment. Egg-hatching rates under treatment with culture filtrates decreased as culture filtrate concentration and exposure time increased relative to that under treatment with SDW. Egg hatching rates were 1.96%, 3%, 16.16%, 26.84%, 40.93%, 52.79%, 45.72%, and 58.26% after 10 days of treatment with 1×, 3×, 5×, 6×, 7×, 9× dilutions, 2216E (medium), and SDW, respectively.

The culture filtrates exhibited different degrees of toxicity against J2 *M. incognita* (Figure 2). The culture filtrates of *P. putida* MCCC 1A00316 exerted nematicidal activity against *M. incognita* even at 9× dilutions. J2 became paralyzed after 24 h of exposure to the culture filtrates of *P. putida* MCCC 1A00316. Nematode mortality suddenly decreased under exposure to 5× and 6× dilutions of the culture filtrates of *P. putida* MCCC 1A00316. Overall, J2 mortalities were proportional to culture filtrate concentrations and exposure duration. These results indicate that *P. putida* MCCC 1A00316 could produce nematicidal extracellular compounds.

### 2.2. Purification of Nematicidal Compounds from Strain MCCC 1A00316

The fermentation supernatant was extracted by using *n*-butyl alcohol in a separatory funnel. The crude extract was visualized through spraying with iodine, and similar fractions were combined. The nematicidal activities of the fractions were tested against *M. incognita* J2. Among the fractions, fraction 1 (F1) provided a retention factor value (Rf) of 0.5 (Appendix A) and exhibited the highest nematicidal activity at the concentrations of 5 and 25 mg/mL (*p* < 0.001) (Figure 3A). 

The further purification of F1 yielded three fractions (F1-1, F1-2, and F1-3), which were then screened through TLC (dichloromethane:methanol 15:1 and iodine spraying) (Appendix A) and again tested against *M. incognita* J2. Exposure to F1-1 at the concentration of 5 and 25 mg/mL resulted in >70% and 100% J2 mortality rates, respectively (Figure 3B). F1-1 was selected for further purification through HPLC. The HPLC chromatogram of fractions 3–27 showed five major peaks at 210 nm at the retention times of 7.479, 8.567, 11.568, 13.346, and 16.796 min (P1–P5) (Figure 4). 

### 2.3. Identification of Purified Nematicidal Compounds

Five compounds were initially purified through HPLC and tested for nematicidal activity against *M. incognita* J2. Results revealed that P2, P3, and P4 had higher nematicidal activity than P1 and P5 (Figure 5). Thus, the structures of P3 and P4 were characterized (the yield of P2 was insufficient for characterization). LC-MS results (Figure 6A,B) showed that the molecular weights of P3 and P4 were 210.1285 and 210.1277, respectively. The two compounds were further identified by comparing their ^1^H- and ^13^C-NMR spectra (Appendix A) with published data.

The peak data of P3 was consistent with those of cyclo(Pro–Ile). The data were as follows: ^1^H-NMR (CD_3_CN, 600 MHz) *δ*:6.46 (1H, s, –NH), 4.06 (1H, t, *J* = 7.9 Hz, H-2), 3.96 (1H, s, H-2′), 3.47 (1H, m, H-5a), 3.39 (1H, m, H-5b), 2.21 (2H, m, H-3), 2.13 (1H, m, H-3′), 1.82-1.92 (2H, m, H-4), 1.39 (1H, m, H-4′a), 1.23 (1H, m, H-4′b), 1.02 (3H, d, *J* = 7.2 Hz, H-6′), and 0.89 (3H, t, *J* = 7.5 Hz, H-5′). ^13^C-NMR (CD_3_CN, 150 MHz) *δ*: 170.8 (C-1), 165.6 (C-1′), 60.4 (C-2), 59.0 (C-2′), 45.2 (C-5), 35.9 (C-3′), 28.7 (C-3), 24.5 (C- 4′), 22.6 (C-4), 15.2 (C-6′), and 12.1 (C-5′). Thus, P3 was identified as cyclo(L-Pro–L-Ile), which was originally reported by Guo et al. [16].

The peak data of P4 was consistent with that of cyclo(Pro–Leu) [19] and were as follows: ^1^H-NMR (CD_3_CN, 600 MHz) *δ*:6.33 (1H, s, –NH), 4.12 (1H, t, *J* = 7.7 Hz, H-2), 4.01 (1H, dd, *J* = 8.6, 3.9 Hz, H-2′), 3.42 (2H, m, H-5), 2.19 (2H, m, H-3), 1.82-1.91 (4H, m, H-4, H-3′), 1.42 (1H, m, H-4′), and 0.94 (3H, d, *J* = 6.5 Hz, H-6′), 0.92 (3H, t, *J* = 6.4 Hz, H-5′). ^13^C-NMR (CD_3_CN, 150 MHz) *δ*: 171.0 (C-1), 167.0 (C-1′), 59.3 (C-2), 53.4 (C-2′), 45.5 (C-5), 38.7 (C-3′), 28.2 (C-3), 24.8 (C- 4′), 23.0 (C-5′), 22.9 (C-4), and 21.3 (C-6′).

### 2.4. Determination of the Absolute Configuration of the Two Cyclodipetides

The acid hydrolysates of cyclodipeptides obtained through treatment with 6 M HCl were treated with Marfey’s FDAA reagent in acetone. The retention time in HPLC of the acid hydrolysates of P4 was compared with that of the FDAA derivatives of individual amino acids to determine the absolute configuration of P4. The retention times of the FDAA derivatives of standard l-Pro, d-Pro, l-Leu, and d-Leu were 11.974, 13.099, 22.403, and 26.963 min, respectively, whereas those of the FDAA derivatives of hydrolysates of P4 were 11.932 and 22.410 min (Figure 7). The above results indicate that compound P4 contained l-Pro and l-Leu.

### 2.5. Anti-M. Incognita Characteristics of Cyclo(l-Pro–l-Leu) from P. putida MCCC 1A00316

The nematicidal effects of the two cyclodipetides from MCCC 1A00316 isolates were evaluated through a direct-contact bioassay. The HPLC-purified cyclo(l-Pro–l-Leu) displayed strong nematicidal activities. J2 mortality reached 42.85% and 63.64% after 48 h of exposure to 67.5 and 135 mg/L cyclo(l-Pro–l-Leu), respectively, and reached 84.30% and 100% after 72 h of exposure to 67.5 and 135 mg/L cyclo(l-Pro–l-Leu), respectively (Figure 8). In addition, egg-hatching rates proportionally decreased as cyclo(l-Pro–l-Leu) concentration increased (Figure 9). Egg-hatching rates of 24.91%, 18.57%, 10.70%, and 9.74% were obtained after 8 days of incubation with 125, 500, 1000, and 2000 mg/mL cyclo(l-Pro–l-Leu). An egg-hatching rate of 53.11% was observed under treatment with SDW. Therefore, an appropriate concentration of cyclo(l-Pro–l-Leu) could be used for the effective control of nematodes by drastically reducing J2 mortality and egg-hatching rates.

The chemotaxis responses of nematodes to cyclo(l-Pro–l-Leu) are presented in Appendix A. The results showed that 2000 and 125 mg/L cyclo(l-Pro–l-Leu) had C.I. values of 0.035 and 0.021 (0 < C.I. <1), respectively, and that 1000 and 500 mg/L cyclo(l-Pro–l-Leu) had C.I. values of −0.014 and −0.048 (−1 < C.I. <0), respectively. Analyzing these data through a Wilcoxon rank–sum test, however, revealed that the number of attracted or repulsed nematodes per side did not show statistically significant differences. These data suggest that different concentrations of cyclo(l-Pro–l-Leu) did not induce chemotactic activity.

## 3. Discussion

We have demonstrated that different concentrations of the culture filtrates of *P. putida* MCCC 1A00316 inhibited the emergence of *M. incognita* from eggs and were lethal to *M. incognita* J2. In addition, *M. incognita* J2 mortality rates and egg-hatching rates were directly proportional to culture filtrate concentration and incubation time. In particular, even 5× dilutions of culture filtrates demonstrated strong nematicidal activity and resulted in *M. incognita* J2 mortality rates of >80% and egg-hatching inhibition rates of <20%. These results suggest that the filtrates contained one or more active compounds that require further characterization. 

We emphasize that our research has several deficiencies. For example, although we obtained satisfactory results for the first experiment, these results lack authenticity. In the analysis of the nematicidal effects of different concentrations of MCCC 1A00316 culture filtrate on *M. incognita* J2 and eggs, we tested 1× and 3×, 5× and 6×, 7× and 9× dilutions of the culture filtrates in different 96/24-well tissue culture plates. The two different filtrates likely affected each other when mixed in the same well. Therefore, the data obtained for the nematicidal activity and egg-hatching effects of culture filtrates with different concentrations may be unreliable. The results of GC-MS analysis revealed that MCCC 1A00316 fermentation broth contained seven nematicidal volatile organic compounds [17].

Because of food and vegetable safety requirements, so the nematicidal activity of the culture filtrates of some strains has been extensively investigated. Lee et al. [20]. reported that the culture filtrates of *Lysobacter capsici* YS1215 had a deleterious effect on the eggs of *M. incognita*. Moreover, lactic acid, a nematicidal compound, had been isolated from the culture filtrate of YS1215. Treatment with the filtrates of the nematophagous fungus *Verticillium leptobactrum* HR43 resulted in the collapse and viability loss of *M. incognita* eggs; these effects suggest that *V. leptobactrum* HR43 produces chitin-degrading enzymes or other active compounds [21]. The results of previous studies are in line with those of our present work, which shows that different concentrations of culture filtrates from microorganisms may demonstrate potential nematicidal activity. Thus, we further investigated the culture filtrate of strain MCCC 1A00316. We were able to detect several nematicidal volatile organic compounds^17^ and separated and purified nematicidal nonvolatile organic compounds.

We isolated and purified a nematicidal compound from the crude extract of *P. putida* MCCC 1A00316. We identified the compound as cyclo(l-Pro–l-Leu) by using various chromatographic techniques, NMR analysis, and Marfey’s method. Cyclo(l-Pro–l-Leu) is a member of the diketopiperazine class of molecules, which includes the smallest cyclic peptides. This compound is synthesized through the head-to-tail folding of linear dipeptides and exhibits an extensive range of biological properties, including antibacterial, antifungal, and antiviral effects [22]. 2,5-diketopiperazine cyclo(l-Pro–l-Leu) from *Streptomyces* sp. has been identified as an antifungal agent with activity against the rice blast fungus *Pyricularia oryzae* and a range of other fungi [23]. Cyclo(l-Pro–l-Leu) produced by the bacterium *Achromobacter xylosoxidans* remarkably inhibits the production of highly toxic, carcinogenic, and teratogenic aflatoxins by the fungus *Aspergillus parasiticus* [24]. It was previously recorded that cyclo(l-Pro–l-Leu) was active against 12 vancomycin-resistant enterococci strains and possesses antileukemic activity [25]. Moreover, cyclo(l-Pro–l-Leu) produced by *P. putida* WCS358 could be used as biosensors for quorum sensing [26]. Interesting, cyclo(d-Pro-l-Leu) purified and identified from *Bacillus amyloliquefaciens* Y1 also has nematicidal activity against *M. incognita* J2 and egg hatching inhibition activity, but the nematicidal activity of cyclo(d-Pro-l-Leu) was significantly lower than that of cyclo(l-Pro–l-Leu) in this study [27]. To our knowledge, we are the first to report that cyclo(l-Pro–l-Leu) isolated from *P. putida* MCCC 1A00316 has nematicidal and egg-hatching inhibition activities. It was previously noted that nematode mortality rate reached 46.19% after 72 h of treatment with 500 mg/L cyclo(l-Ile–l-Pro) in our former study [16], whereas, in this study, it was reached 84.30% after 72 h of treatment with 67.5 mg/L cyclo(l-Pro–l-Leu). However, different concentrations of cyclo(l-Pro–l-Leu) failed to elicit chemotactic activity from nematodes. Therefore, it was confirmed that *P. putida* MCCC 1A00316 and its metabolites cyclic dipeptides displayed potential as biocontrol agents against nematodes.

## 4. Materials and Methods

### 4.1. Collection of M. Incognita Eggs and J2

*M. incognita* eggs were collected from the roots of infested tomato plants (*Solanum lycopersicum* L.) that had been previously infected with nematodes. The plants were maintained under greenhouse conditions at 23–26 °C and relative humidity of 40–60%. The plants were watered manually once a day. After 45 days, the plants were uprooted, and their roots were rinsed free of soil with tap water. Egg masses were transferred to a bottle by using a dissecting needle and shaken with 1% NaOCl solution for 3 min. The egg mass suspension was then passed through a series of filters with pore sizes of 74, 45, and 25 μm. Sterilized eggs that were retained on the 25 μm filter were sprayed with sterile distilled water (SDW) for collection. J2 were obtained under sterile conditions by using a modified Baermann funnel method [28,29].

### 4.2. Preparation of the Fermentation Supernatant of P. putida MCCC 1A00316 

Strain MCCC 1A00316 was isolated from Antarctic soil and identified as *P. putida* on the basis of 16S rDNA sequence homology and physiological and biochemical characteristics [19]. The strain was cultured in 30 mL flasks containing 15 mL of 2216E broth [30]. The 2216E broth was prepared with 10 g of peptone, 5 g of yeast powder, 1 g of beef extract, 0.1 g of ferric citrate, 1 g of sodium acetate, 19.45 g of NaCl, 0.75 g of MgCl_2_, 0.75 g of MgSO_4_, 1 g of CaCl_2_, 0.55 g of KCl, 0.16 g of NaHCO_3_, 0.08 g of KBr, 34 mg of SrCl_2_, 22 mg of H_3_BO_3_, 4 mg of Na_2_SiO_3_, 2.4 mg of NaF, 8 mg of Na_2_HPO_4_, 0.5 mg of MnCl_2_, 0.5 mg of CuSO_4_, and 10 mg of ZnSO_4_. The pH and temperature of the seed medium were maintained at 7.6–7.8 and 28 °C, respectively. The medium was shaken at 180 rpm. After 18 h, 1% seed liquid (2.5 mL) was transferred into a 500 mL flask containing 250 mL of 2216E medium. The flask was then shaken for 48 h at 28 °C at 180 rpm. The fermentation culture was centrifuged at 8000 r/min for 10 min at 4 °C to obtain the supernatant. Filtrates were passed through a 0.22 μm filter to remove bacterial cells and were used at the original concentration or to prepare 1×, 3×, 5×, 6×, 7×, 9× dilutions with SDW.

### 4.3. Effect of the Fermentation Supernatant of P. putida MCCC 1A00316 on Egg-Hatching

*M. incognita* egg suspensions (10 μL) were prepared (180 eggs/well) as described, transferred to separate wells of a 24-well tissue culture plate, and mixed with 500 μL of culture filtrates with various concentrations. Next, 4 μL (10 mg/mL) of chloramphenicol was added to each well to inhibit bacterial growth. Then, 1× and 3×, 5× and 6×, 7× and 9× dilutions of culture filtrates were transferred to different 24-well tissue culture plates. 2216E medium and SDW were used as controls. Experiments were performed three times, and every treatment was replicated three times. Plate lids were sealed with parafilm, and the plates were maintained at 28 °C. An inverted microscope (XDS-1B COIC, Chongqing Mike Photoelectric Instrument Limited Company, Chongqing, China) was used to observe egg-hatching rates after 2, 4, 6, 8, and 10 days of incubation. 

### 4.4. Effects of Strain MCCC 1A00316 Culture Filtrates on M. incognita J2

*M. incognita* J2 were suspended in SDW. The J2 suspensions (3 μL) were transferred to the wells of a 96-well tissue culture plate (50 J2/well) containing 200 μL of different dilutions of culture filtrates. A total of 3 μL (10 mg/mL) of chlorampheniol was added to the wells to inhibit bacterial growth. Next, 1× and 3×, 5× and 6×, 7× and 9× dilutions of culture filtrates were placed in different 96-well tissue culture plates. 2216E medium and SDW were used as controls. The experiments were performed three times, and every treatment was replicated three times. Plate lids were sealed with parafilm, and the plates were maintained at 28 °C. Juveniles were observed with the aid of an inverted microscope (XDS-1B COIC) under 10× and 40× after 24, 48, and 72 h and classified as motile or immotile/paralyzed.

### 4.5. Crude Extraction of Nematicidal Compounds

The procedure used to isolate pure compounds from strain MCCC 1A00316 is illustrated in Figure 10. The fermentation supernatant of *P. putida* MCCC 1A00316 was mixed with equal volumes (250 mL) of n-butyl alcohol through 3 min of shaking in a separatory funnel. The extracted organic phases were evaporated to dryness at 58 °C by rotary evaporation, dissolved with 2 mL of methanol, and filtered using the funnel filter paper method. 

### 4.6. Purification of Two Cyclodipetide Compounds

Filtrate dissolved in methanol was mixed with a small quantity of silica gel (200–300 mesh) and then evaporated to dryness at 58 °C. The dry powder was eluted through a big silica gel column (60 mm × 53 mm × 457 mm) containing 170 g of silica gel (200–300 mesh). Elution was performed by using 400 mL of ethyl acetate:methanol at varying ratios (12:1 [F1], 8:1 [F2], 5:1 [F3], or 2:1 [F4]). Fractions were monitored through thin-layer chromatography (TLC) on silica gel plates (F254, 10 cm × 20 cm) with ethyl acetate:methanol (6:1, *v*/*v*). Bands were visualized through spraying with iodine. Similar fractions were combined and tested against J2. The fraction that showed high nematicidal activity was loaded on a small silica gel column (32 mm × 26 mm × 305 mm) containing 45 g of silica gel (60–80 mesh), eluted with various ratios (80:1 [F1-1], 50:1 [F1-2], or 30:1 [F1-3]) of dichloromethane:methanol, screened through TLC (dichloromethane:methanol 15:1 and spraying with iodine), and tested against J2. 

### 4.7. High-Performance Liquid Chromatography 

The fraction exhibiting the most intense nematicidal activity was freeze-dried and then dissolved in ultrapure water. The dissolved fraction was filtered through a 0.22 μm filter. Next, 100 μL samples were loaded for high-performance liquid chromatography (HPLC) analysis by injection onto an semi-preparative chromatographic column (Ultimate XB, 10 μm, 10 mm × 250 mm, Welch, shanghai, China). Acetonitrile:water (20:80, *v*/*v*) was used as the mobile phase. A variable-wavelength recorder set at 210 nm was applied to detect the compounds eluted from the column at a flow rate of 3 mL/min [17]. Five peaks were collected. Five peak samples were collected separately. The purity of 10 μL samples was tested by using a high-performance liquid chromatography (HPLC) system equipped with a analytical column (TC-C18 column, 5 μm, 4.6 mm × 250 mm, Agilent Technologies, Santa Clara, CA, USA). The nematicidal activities of the compounds corresponding to the five peaks were tested, and the compounds corresponding to three peaks with high nematicidal activity were selected for structural analysis.

### 4.8. Liquid Chromatography Time-of-Flight Mass Spectrometry 

The three compounds with intense nematicidal activity were analyzed through liquid chromatography time-of-flight mass spectrometry (Agilent Technologies 6540UHD Accurate-Mass Q-TOF LC-MS). Next, 1 μL samples were analyzed via reverse-phase HPLC (C18 column, particle size 3.5 μm, 2.1 mm × 150 mm, Agilent Technologies) with a gradient of acetonitrile:water (20:80, *v*/*v*) at a flow rate of 0.3 mL/min. The HPLC eluate was introduced into a mass spectrometer via an ESI interface at a spray voltage of 4.0 kV. The mass spectrometer was operated in positive-ion mode with a capillary temperature of 325 °C and a capillary voltage of 130 V. Mass spectra were acquired by scanning the mass range from *m*/*z* 100 to *m*/*z* 1700.

### 4.9. Identification of Nematicidal Compounds through Nuclear Magnetic Resonance Spectroscopy 

The structures of the compounds were determined by using a DRX 600 NMR instrument (Bruker Karlsruhe, Germany) equipped with a 5 mm microprobe. Approximately 5 mg of the purified compound was dissolved in acetonitrile-d3 (C2D3N) and then subjected to spectral analysis. NMR spectra were recorded on the instrument, which was operated at room temperature and at 600 and 125 MHz for ^1^H-NMR and ^13^C-NMR, respectively. Chemical shifts are reported in ppm (d) by using C2D3N as the solvent unless otherwise indicated.

### 4.10. Determination of the Absolute Configuration of Compounds 

A solution of three different compounds (1.5 mg) in 6 M HCl (1 mL) was heated to 120 °C for 24 h. The solution was then evaporated to dryness. The residue was redissolved in H_2_O (100 μL), transferred to a 1 mL reaction vial, and treated with a 2% solution of FDAA (200 μL) in acetone and then with 1.0 M NaHCO_3_ (40 μL). The reaction mixture was heated at 45 °C for 1 h, cooled to room temperature, and acidified with 2.0 M HCl (20 μL). In a similar fashion, standard d- and l-amino acids were derivatized separately. The derivatives of the hydrolysates and standard amino acids were subjected to HPLC analysis (SunFire LC-20AD, C18 column; 5 μm, 4.6 mm × 250 mm; 1.0 mL/min, Waters, Milford, CT, USA) at 30 °C with the following gradient program: solvent A, water + 0.2% TFA; solvent B, MeCN; linear gradient 0 min 25% B, 40 min 60% B, 45 min 100% B; and UV detection at 340 nm [31].

### 4.11. Nematicidal Activity of Cyclodipetides from Strain MCCC 1A00316

The hatched J2 of *M. incognita* were suspended in SDW. J2 suspensions (3 μL) were added to the wells of a 96-well tissue culture plate (50 J2/well) containing 100 μL of cyclodipetide solutions with different concentrations. SDW was used as the control. The experiments were performed three times, and every treatment was replicated three times. Plate lids were sealed with Parafilm. The plates were maintained at 28 °C. Juveniles were observed with the aid of an inverted microscope (XDS-1B COIC) under 10× and 40× magnification after 24, 48, and 72 h of exposure and categorized as motile or immotile/paralyzed.

### 4.12. Effect of Cyclo(l-Pro–l-Leu) on the Egg-Hatching of M. incognita

The influence of the purified compound on the egg-hatching rates of *M. incognita* was examined in a 96-well microtiter plate containing 60 eggs. Various concentrations of the purified compounds in SDW were added to each well to the final concentrations of 125, 500, 1000, and 2000 mg/L. SDW was used as the control. The number of juveniles that hatched from eggs at 2, 5, and 8 days of incubation was counted under an inverted microscope.

### 4.13. Chemotaxis of M. incognita J2 to cyclo(l-Pro–l-Leu)

The chemotaxis assay was performed in accordance with the method of Zhai et al. [17]. A 5 mm filter paper disc that had been immersed in various concentrations of cyclo(l-Pro–l-Leu) was placed on one side (area A) of a 35 mm Petri dishes containing 2% water agar. A filter paper immersed in SDW was placed on the the opposite side of the plate (area B) as the control. Subsequently, 100 J2 *M. incognita* were added to the center (area D) of the Petri dish. The dish was then incubated in a dark cabinet at 20 °C for 8 h. The numbers of *M. incognita* J2 in areas A and B were then counted under a dissecting microscope for the calculation of the chemotaxis index (C.I.) [32], which is defined as follows:
C.I. = (the number of nematodes in the test area − the number of nematodes in the control area)/(the number of nematodes in test area + the number of nematodes in the control area).(1)

*M. incognita* J2 was considered attracted to the tested sample if 0 < C.I. < 1, repelled if −1< C.I. < 0, and unaffected if C.I. = 0. Experiments were performed in triplicate, and treatments were replicated three times.

### 4.14. Data Analysis and Statistics

The mortality rates of *M. incognita* in in vitro bioassays were corrected by Abbott’s formula. Data from all assays were analyzed through by one-way analysis of variance with SPSS 20 (IBM, Armonk, NY, USA). Data from the chemotaxis assay were analyzed through a homogeneity test of variance. Paired Student’s *t*-test was performed if variance was homogeneous (*p* ≤ 0.05). Otherwise, the Wilcoxon rank–sum test was used. Duncan’s multiple range test was employed to test for significant differences among the anti-*M. incognita* activities of the culture filtrates of strain MCCC 1A00316. Statistical comparisons between two experimental values were performed via *t*-test, and significant differences were determined in accordance with a threshold of * *p* < 0.05; ** *p* < 0.01; *** *p* < 0.001.

## Figures and Tables

**Figure 1 molecules-24-00768-f001:**
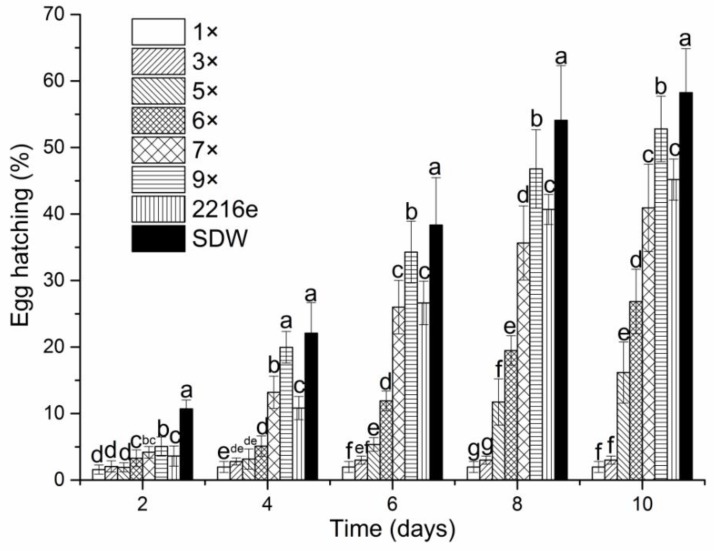
Inhibitory effects of the culture filtrates of strain MCCC 1A00316 on the hatching of *M. incognita* eggs after 2, 4, 6, 8 and 10 days of exposure. Each value represents the mean ± standard error with six replicates each. Means (±SEs) for each treatment concentration followed by the same letters within bars are not significantly different (*p* = 0.05) as revealed through Duncan’s test.

**Figure 2 molecules-24-00768-f002:**
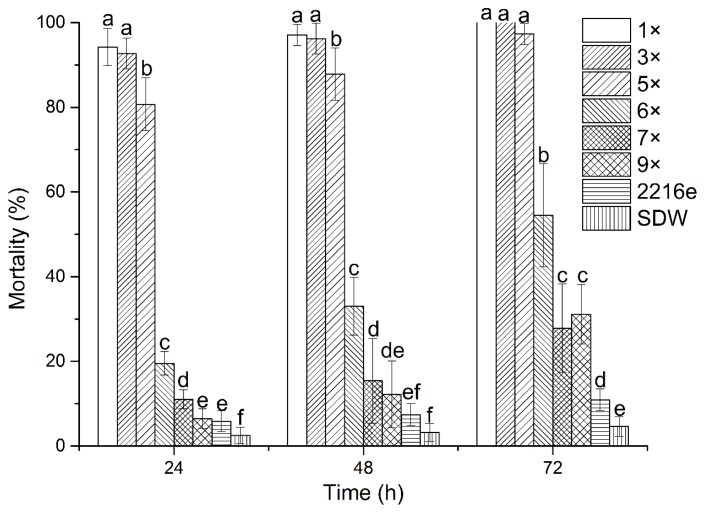
Effects of the culture filtrates of strain MCCC 1A00316 on the mortality of *M. incognita* J2 at 24, 48, and 72 h of exposure. Each value represents the mean ± standard error with six replicates each. Means (±SEs) for each treatment concentration followed by the same letters within bars are not significantly different (*p* = 0.05) as revealed through Duncan’s test.

**Figure 3 molecules-24-00768-f003:**
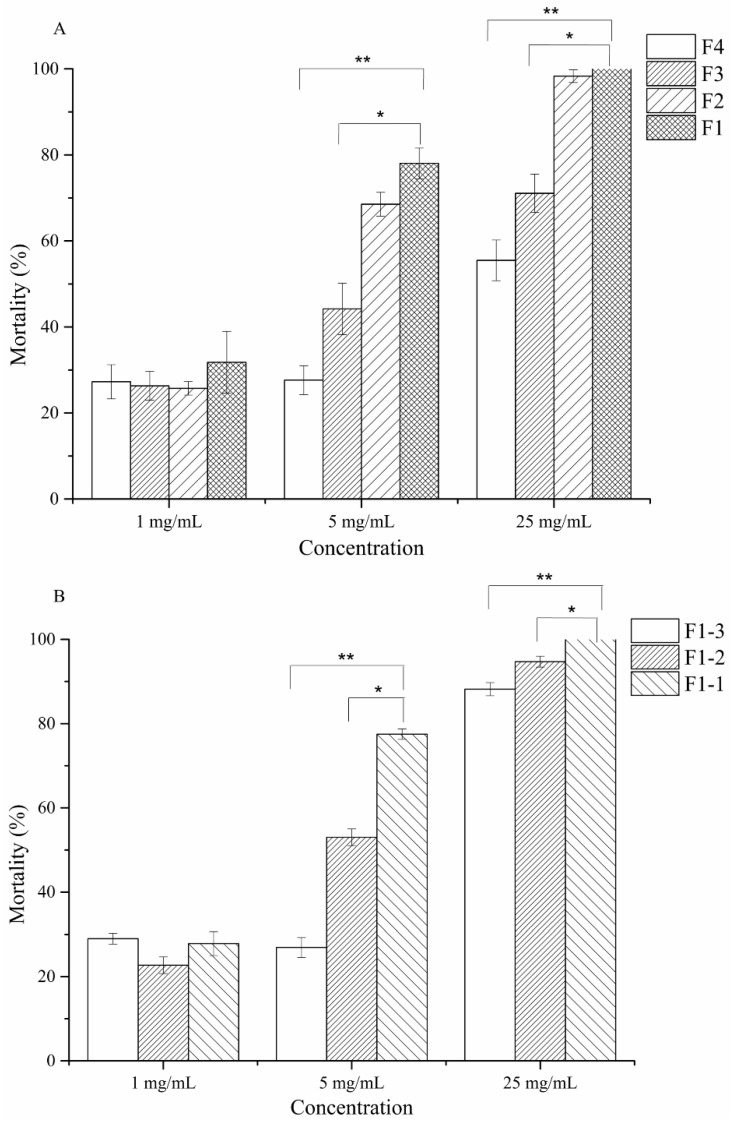
Nematicidal activity of the four fractions obtained from the strain MCCC 1A00316 fermentation supernatant through big silica gel chromatography by using ethyl acetate/methanol (12:1 [F1], 8:1 [F2], 5:1 [F3], 2:1 [F4]) (**A**). nematicidal activity of the three fractions obtained from F1 through small silica gel chromatography by using dichloromethane/methanol (80:1 [F1-1], 50:1 [F1-2], 30:1 [F1-3]). (**B**). Values are mean ± SD. * indicates a difference and ** indicates a statistically significant difference as determined through Student’s *t*-test (*p* < 0.05).

**Figure 4 molecules-24-00768-f004:**
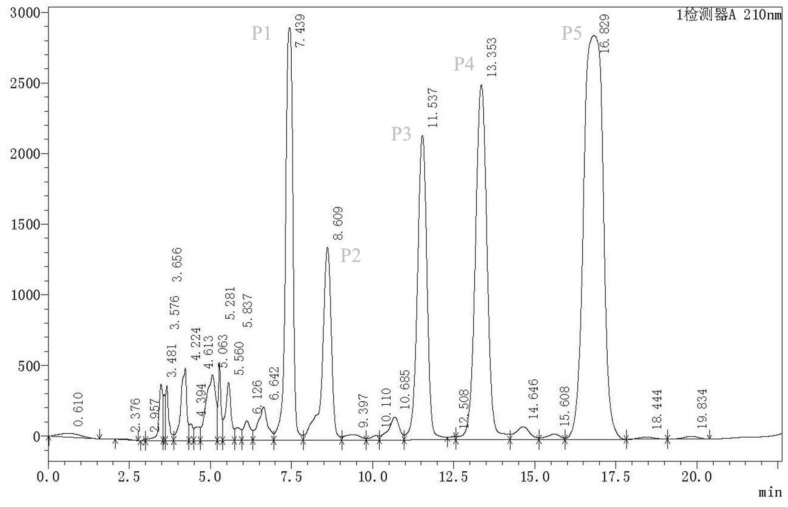
HPLC profiles of P1–P5 purified from the fermentation supernatant of strain MCCC 1A00316. As shown in the figure, P1–P5 are indicated in7.479, 8.567, 11.568, 13.346, and 16.796 min, respectively.

**Figure 5 molecules-24-00768-f005:**
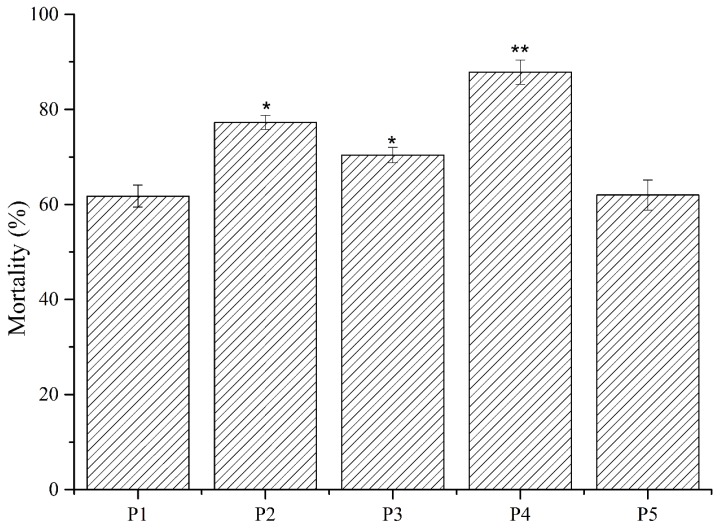
Nematicidal activity of P1–P5 obtained from the fermentation supernatant of strain MCCC 1A00316. Values are mean ± SD. * indicates a difference and ** indicates a statistically significant difference as determined through Student’s *t*-test (*p* < 0.05).

**Figure 6 molecules-24-00768-f006:**
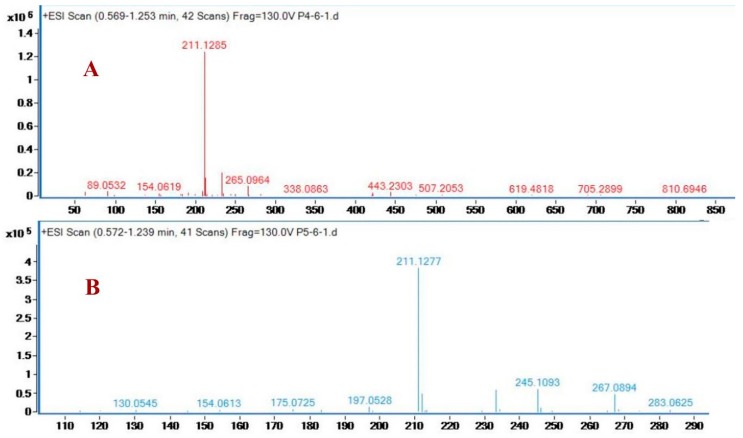
LC-MS chromatogram of P3 (**A**) and P4 (**B**).

**Figure 7 molecules-24-00768-f007:**
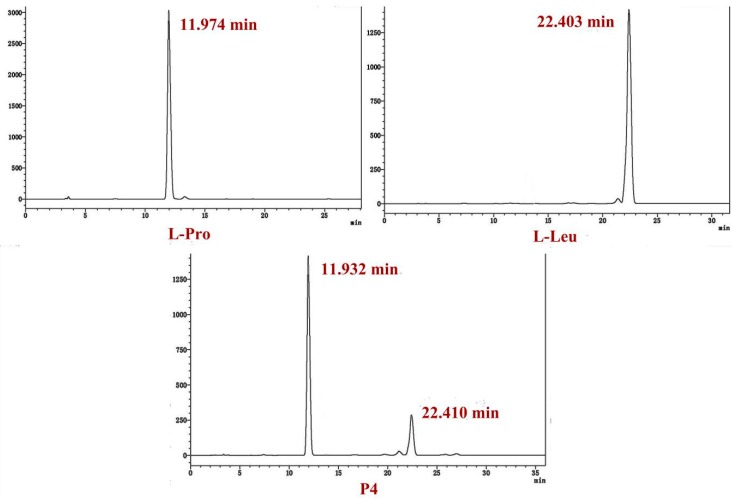
HPLC profile of FDAA derivatives of l-Pro, l-Leu, and P4.

**Figure 8 molecules-24-00768-f008:**
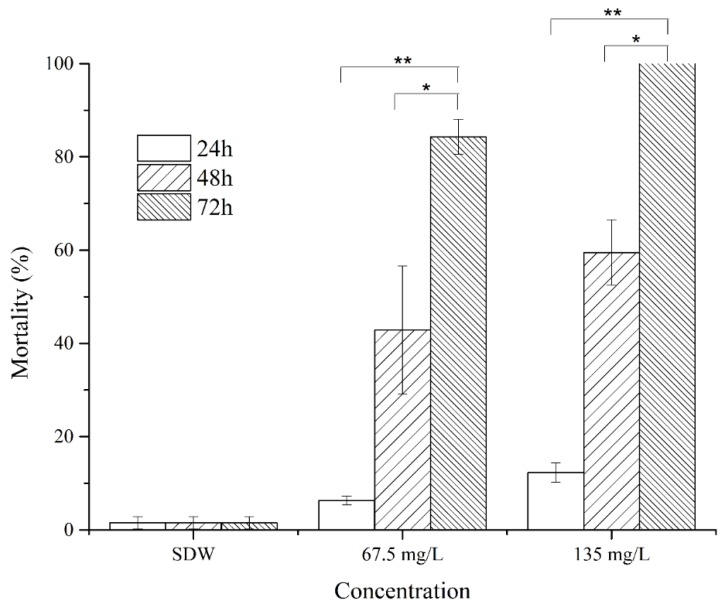
Nematicidal activity of cyclo(l-Pro–l-Leu) obtained from the fermentation supernatant of strain MCCC 1A00316. Values are mean ± SD. * indicates a difference and ** indicates a statistically significant difference as determined through Student’s *t*-test (*p* < 0.05).

**Figure 9 molecules-24-00768-f009:**
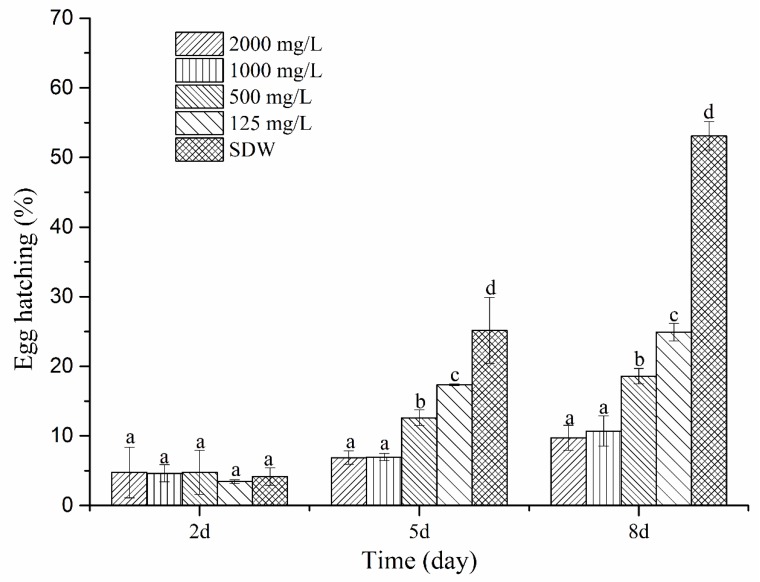
Effect of cyclo(l-Pro–l-Leu) purified from *P. putida* MCCC 1A00316 culture on egg hatching. Each value represents the mean ± standard error with six replicates each. Means (±SEs) for each treatment concentration followed by the same letters within bars are not significantly different (*p* = 0.05) as revealed through Duncan’s test.

**Figure 10 molecules-24-00768-f010:**
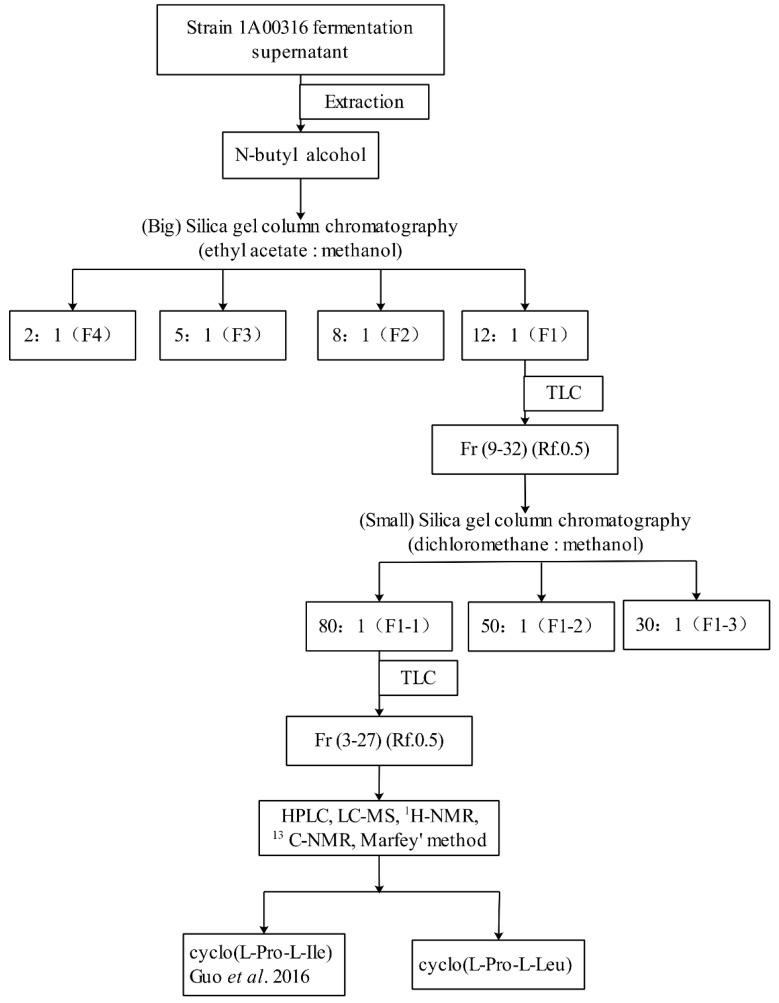
Diagram of the separation and purification of two cyclodipetides from strain MCCC 1A00316.

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
