# Peer review of "Cyclo(l-Pro–l-Leu) of Pseudomonas putida MCCC 1A00316 Isolated from Antarctic Soil: Identification and Characterization of Activity against Meloidogyne incognita"

_molecules, 2019, doi:10.3390/molecules24040768_

Round 1
Reviewer 1 Report
The authors describe the isolation and identification of a dipeptide with apparent nematicidal activity from a previously isolated strain of Pseudomonas. Although in general the work is well described, there is lots of room for improvement.
Specifically, it should be emphasized what is new in this study compared to Guo et al. (2016). I understand that the strain was isolated previously, but were there new methods introduced or existing methods improved etc.
There are some other important existing publications about cyclic dipeptides that have not even been mentioned in this MS and should be included in comparison and discussion:
Jamal et al. (2017) Molecules, 22:1839. cyclo(D-Pro-L-Pro) nematocide
Degrassi et al. (2002) Curr. Microbiol. 45:250-254. four cyclic peptides
Other points to address:
P1-L16/17 "larval mortality end egg-hatching rates...are directly proportional to the concentration"
I believe the authors mean the inhibition of the egg-hatching rate and not egg-hatching rate per se as this would suggest that higher concentrations of the nematocide lead to highre hatching rates.
P2-L52 "pathogenic bacterial and fungal species..." pathogenic to what, please clarify
P2-L82/83 This is very confusing for the reader as abbreviations are only explained in the following section (Materials and Methods). It should be mentioned here what SDW and 2216E stand for. Also I read "treatment with 1x, 3x, 5x etc.) as concentrations and not as dilutions as the authors intend. This should be clarified and also mentioned in the figure legends.
P2-L84 again this statement is misleading as it seems to refer to different concentrations (dilutions) rather than different culture filtrates
P4-L106 clarify what is "further purification"
P4-Fig 3 what does "concentration" mg/ml refer to and how was it measured ?
P5-Fig 4 P1-P5 are not defined, it should be clarified in text that it refers to the peaks of the chromatogram, and it should also be indicated in the chromatogram itself.
P6-Fig 5 How where the different peaks standardized in terms of e.g. concentration ? How do they relate to initially tested fractions and concentrations/dilutions.
P7-Fig 7 bad quality and hardly readable. What about commercially available standards, they could have been spiked to identify peaks rather than by comparison of retention times only.
P10-L224 ff
Comparison needs to be done with the results obtained by Jamal et al 2017 see above. Also it would be good to confirm the results with commercially available dipeptides and compare them to the 3 possible isomeres in terms of activity and required concentrations.
Author Response
Response to Reviewer 1 Comments
Point 1: Specifically, it should be emphasized what is new in this study compared to Guo et al. (2016). I understand that the strain was isolated previously, but were there new methods introduced or existing methods improved etc.
Response 1: Thank you for your careful comments. In our experiments, methods of separation, purification and identification of small molecular compounds are consistent with Guo et al. (2016). However, there are some new methods and results for the Cyclo(L-Pro–L-Leu) characterization analysis in our study. For example, the methods for the inhibitory ability test of the cyclodipeptide on the hatching of M. incognita eggs, and the chemotaxis activity of M. incognita J2 are different from the Guo paper.
Point 2: There are some other important existing publications about cyclic dipeptides that have not even been mentioned in this MS and should be included in comparison and discussion:
Jamal et al. (2017) Molecules, 22:1839. cyclo(D-Pro-L-Pro) nematocide
Degrassi et al. (2002) Curr. Microbiol. 45:250-254. four cyclic peptides
Response 2: Thank you very much for your valuable suggestion. We have read two papers carefully and compare them with our study in the discussion section.
Point 3: P1-L16/17 "larval mortality end egg-hatching rates...are directly proportional to the concentration"
I believe the authors mean the inhibition of the egg-hatching rate and not egg-hatching rate per se as this would suggest that higher concentrations of the nematocide lead to highre hatching rates.
Response 3: Thank you very much for your valuable advice. We will use “The larval mortality and egg-hatching inhibition rates of M. incognita will increase with the rising concentration of culture filtrates of P. putida 1A00316 and the duration of exposure” instead of "larval mortality end egg-hatching rates...are directly proportional to the concentration".
Point 4: P2-L52 "pathogenic bacterial and fungal species..." pathogenic to what, please clarify
Response 4: Thank you for your careful comments. The pathogenic contains Pseudomonas aeruginosa, Pasteuria penetrans and Bacillus cereus.
Point 5: P2-L82/83 This is very confusing for the reader as abbreviations are only explained in the following section (Materials and Methods). It should be mentioned here what SDW and 2216E stand for. Also I read "treatment with 1x, 3x, 5x etc.) as concentrations and not as dilutions as the authors intend. This should be clarified and also mentioned in the figure legends.
Response 5: Thank you for your careful reading of our manuscript. In the line 82 and 83, we have revised the problems of SDW and 2216E in the new manuscript. In L255-256, we did make a mistake, so we modify it again by using 1×, 3×, 5×, 6×, 7×, 9× dilutions and make the dilutions group clear.
Point 6: P2-L84 again this statement is misleading as it seems to refer to different concentrations (dilutions) rather than different culture filtrates
Response 6: Thank you for your careful reading of our manuscript. In the line 84, the culture filtrates means different concentrations dilutions (3×, 5×, 6×, 7×, 9× culture filtrates were diluted with sterile distilled water).
Point 7: P4-L106 clarify what is "further purification"
Response 7: Thank you for your careful reading of our manuscript. In the line 106, The further purification of F1 is that F1 fraction showed the highest nematicidal activity was loaded on a small silica gel column (32 mm × 26 mm × 305 mm) containing 45 g of silica gel (60–80 mesh) eluted with various ratios (80:1 [F1-1], 50:1 [F1-2], or 30:1 [F1-3]) of dichloromethane : methanol.
Point 8: P4-Fig 3 what does "concentration" mg/ml refer to and how was it measured ?
Response 8: Thank you for your careful reading of our manuscript. In the Fig 3, the concentration is the concentration of crude extract from silica gel column chromatography eluant. The concentration is determined by first weighing the mass of the centrifuge tube, adding corresponding fractions, freeze-drying, weighing again, adding a certain amount of water to dissolve. The two weighing results are subtracted and then divided by the water volume, and finally, so we get the "concentration" mg/mL.
Point 9: P5-Fig 4 P1-P5 are not defined, it should be clarified in text that it refers to the peaks of the chromatogram, and it should also be indicated in the chromatogram itself.
Response 9: Thank you very much for your valuable suggestion. According with your advice, we amend the Fig 4 part and figure 4 legends. We add some captions in Fig 4, figure 4 legends and in text.
Point 10: P6-Fig 5 How where the different peaks standardized in terms of e.g. concentration ? How do they relate to initially tested fractions and concentrations/dilutions.
Response 10: Thank you for your careful reading of our manuscript. Firstly, we collected different peak samples by HPLC, then freeze-dried them, weighed them, and added a certain amount of water to make each peak sample have the same final concentration. Initial dilution experiments showed that 1A00316 fermentation broth contained compounds that killed the larva of M. incognita or inhibited egg hatching of M. incognita. Under this logical basis, we next used chromatography and chromatography to isolate and identify what these compounds were.
Point 11: P7-Fig 7 bad quality and hardly readable. What about commercially available standards, they could have been spiked to identify peaks rather than by comparison of retention times only.
Response 11: We are grateful for your suggestion. According to your comment, we have improved the resolution of the picture. There are many methods to identify the absolute structure of compounds. Marfey’s method was discovered by Marfey in 1984, and this method has been verified in many later literatures, for example “Purification of an antifungal compound, cyclo(l-Pro-d-Leu) for cereals produced by Bacillus cereus subsp. thuringiensis associated with entomopathogenic nematode”.
Point 12: P10-L224 ff
Comparison needs to be done with the results obtained by Jamal et al 2017 see above. Also it would be good to confirm the results with commercially available dipeptides and compare them to the 3 possible isomeres in terms of activity and required concentrations.
Response 12: Thank you very much for your valuable suggestion. We have read these two papers carefully and compare them in the discussion section. Thank you very much for your suggestion again. Since every step of our experiment has been strictly analyzed and identified, we strongly believe that the separated and purified compound is the same as the commercial compound. For the other three types of cyclic dipeptides, we will compare their activity in later work.

Reviewer 2 Report
he manuscript presents cyclo(L-Pro–L-Leu) of Pseudomonas putida 1A00316 isolated from Antarctic soil: identification and characterization of activity against Meloidogyne incognita . The manuscript is presented in a lucid manner, yet some points need to be further explained or revised:
• There is no information about validation of chromatographic methods (section 4.7. and 4.8.). How did authors inspect the purity of peaks in HPLC analysis?
• HPLC chromatogram (Figure 4) and its caption are badly described and unclear.
• Caption to Figure 6 – “chromatograph” should be “chromatogram”.
• Figure 7 is of poor quality.
• What was injection volume of the samples during HPLC analysis(section 4.7. and 4.8.)?
• The section "Conclusion" is missing.
Author Response
Response to Reviewer 2 Comments
Point 1: There is no information about validation of chromatographic methods (section 4.7. and 4.8.). How did authors inspect the purity of peaks in HPLC analysis?
Response 1: Thank you very much for your valuable suggestion. The chromatographic method we used was mainly referred to “Comparative genomic and functional analyses: unearthing the diversity and specificity of nematicidal factors in Pseudomonas putida strain 1A00316”. I'm really sorry that we did miss some information, but we have modified section 4.7 in the new manuscript. Regarding the verification peak sample purity, we first purified the crude extract from the previous step with a semi-preparative column, and then tested the purity of each peak sample with an analytical column. Of course, the results of LC-MS also indicate that the sample has high purity.
Point 2: HPLC chromatogram (Figure 4) and its caption are badly described and unclear.
Response 2: Thank you very much for your valuable suggestion. According with your advice, we amend the Fig 4 part and figure 4 legends. We add some captions in Fig 4 and figure 4 legends.
Point 3: Caption to Figure 6 – “chromatograph” should be “chromatogram”.
Response 3: Thank you very much for your valuable suggestion. According with your advice, we amend the caption of Figure 6.
Point 4: Figure 7 is of poor quality.
Response 4: We are grateful for your suggestion. According to your comment, we have improved the resolution of the picture.
Point 5: What was injection volume of the samples during HPLC analysis(section 4.7. and 4.8.)?
Response 5: Thank you for your careful reading of our manuscript. The injection volume of the peak sample purified by semi-prepared isolates was 100 microliters, and the injection volume of the peak sample identified by analytical column was 10 microliters.
Point 6: The section "Conclusion" is missing.
Response 6: Thank you very much for your valuable suggestion. According to the guidance to authors in the journal, this conclusion section is not mandatory, but can be added to the manuscript if the discussion is unusually long or complex. So our conclusion section is merged in the last paragraph of the discussion section.

Round 2
Reviewer 1 Report
I think the MS has improved in terms of clarity, and most of my concerns have been addressed.
There are still a few things that should be changed before publications:
Fig. 7 although it seems to have improved in resolution it might still not meet the criteria for publication (please check with editor).
page 12, lines 231-248: Please check the whole paragraph for spelling and grammar errors. It seems to me that text has rather hastily been added. A few sentences don't make sense and there are repeated words. It needs to be revised before publication !!
Author Response
Point 1: Fig. 7 although it seems to have improved in resolution it might still not meet the criteria for publication (please check with editor).
Response 1: Thank you for your careful comments. According to your comment, we have improved the resolution of the picture. In the process of modification, we find an error that P4 compound rather than P5 compound was identified in the absolute conformation. We have modified it in the new version.
Point 2: page 12, lines 231-248: Please check the whole paragraph for spelling and grammar errors. It seems to me that text has rather hastily been added. A few sentences don't make sense and there are repeated words. It needs to be revised before publication !!
Response 2: Thank you very much for your valuable suggestion. We have checked lines 231-248 carefully, and we did find some mistakes and language errors. We have made some modifications in the new version.

Reviewer 2 Report
Authors improved their manuscript according to reviewer's comments.
Author Response
Point 1: Authors improved their manuscript according to reviewer's comments.
Response 1: Thank you very much for your valuable comments.
